# The Evolutionary Significance of RNAi in the Fungal Kingdom

**DOI:** 10.3390/ijms21249348

**Published:** 2020-12-08

**Authors:** Carlos Lax, Ghizlane Tahiri, José Alberto Patiño-Medina, José T. Cánovas-Márquez, José A. Pérez-Ruiz, Macario Osorio-Concepción, Eusebio Navarro, Silvia Calo

**Affiliations:** 1Department of Genetics and Microbiology, Faculty of Biology, University of Murcia, 30100 Murcia, Spain; carlos.lax@um.es (C.L.); ghizlane.tahiri@um.es (G.T.); josetomas.canovas@um.es (J.T.C.-M.); joseantonio.perez6@um.es (J.A.P.-R.); macario.osorio@um.es (M.O.-C.); sebi@um.es (E.N.); 2Instituto de Investigaciones Químico Biológicas, Universidad Michoacana de San Nicolás de Hidalgo, Ciudad Universitaria, Morelia, Michoacán CP 58030, Mexico; jpatino@umich.mx; 3School of Natural and Exact Sciences, Pontificia Universidad Católica Madre y Maestra, 51033 Santiago de los Caballeros, Dominican Republic

**Keywords:** RNAi, fungi, Dicer, Argonaute, RNA-dependent RNA polymerase, RNA silencing, siRNA

## Abstract

RNA interference (RNAi) was discovered at the end of last millennium, changing the way scientists understood regulation of gene expression. Within the following two decades, a variety of different RNAi mechanisms were found in eukaryotes, reflecting the evolutive diversity that RNAi entails. The essential silencing mechanism consists of an RNase III enzyme called Dicer that cleaves double-stranded RNA (dsRNA) generating small interfering RNAs (siRNAs), a hallmark of RNAi. These siRNAs are loaded into the RNA-induced silencing complex (RISC) triggering the cleavage of complementary messenger RNAs by the Argonaute protein, the main component of the complex. Consequently, the expression of target genes is silenced. This mechanism has been thoroughly studied in fungi due to their proximity to the animal phylum and the conservation of the RNAi mechanism from lower to higher eukaryotes. However, the role and even the presence of RNAi differ across the fungal kingdom, as it has evolved adapting to the particularities and needs of each species. Fungi have exploited RNAi to regulate a variety of cell activities as different as defense against exogenous and potentially harmful DNA, genome integrity, development, drug tolerance, or virulence. This pathway has offered versatility to fungi through evolution, favoring the enormous diversity this kingdom comprises.

## 1. Introduction

RNA interference (RNAi) or RNA silencing has been deeply studied in the last two decades, as its discovery entailed a revolution in the understanding of the regulation of gene expression. This RNAi pathway, broadly conserved in eukaryotes (Table 1), uses small interfering RNAs (siRNAs) to suppress gene expression of homologous sequences. These siRNAs, of 20–30 nucleotides (nt) long, are produced from double-stranded RNA (dsRNA) by an RNase III called Dicer (Dcr) and loaded into an RNA-induced silencing complex (RISC), which contains an Argonaute protein (Ago) that drives the selective degradation of homologous messenger RNAs (mRNA), as well as translational or transcriptional repression of target sequences. Moreover, in fungi and other organisms, an RNA-dependent RNA polymerase (Rdp or RdRP) generates dsRNA from certain single-stranded RNA (ssRNA) or from the target messenger RNA, activating or amplifying the silencing response, respectively [1,2].

Fungi have proven to be excellent model organisms for the study of the RNAi pathway, since many of the discoveries accomplished in these organisms were later extended to higher eukaryotes. In fact, one of the first RNA silencing phenomena reported was found in *Neurospora crassa*, which is an essential model organism to study modern genetics. *N. crassa* has developed different RNAi mechanisms, but the two that were originally found are the best described. The first is called quelling, a post-transcriptional gene silencing (PTGS) guided by siRNAs [15], which suppresses transposons and virus infections [16]. Quelling is triggered by the introduction of transgenes homologous to an endogenous gene. After the transgene is transcribed, it follows a canonical pathway to activate silencing, carried out by an Rdp protein (QDE-1), two Dicer-like proteins (Dcl1 or Dcl2), and an Argonaute (QDE-2). The second mechanism is called meiotic silencing of unpaired DNA (MSUD) and is involved in silencing genes that are not paired with their partner on the homologous chromosome during meiosis [17]. This mechanism is present not only in *N. crassa* but also in other ascomycetes, such as *Gibberella zeae,* and operates during prophase I [18,19]. Some of the elements involved in the canonical RNAi pathway, such as Dcl1, are necessary for this other mechanism, as well as a MSUD-specific Rdp (SAD-1), a second Argonaute (SMS-2), and the helicase SAD-3. These proteins form a multiprotein complex located at the perinuclear region that acts generating MSUD-associated siRNAs (masiRNAs) [20]. After these first discoveries in *N. crassa*, the RNAi mechanism was found in several other fungi, such as *Schizosaccharomyces pombe* [21], *Cryptococcus* [22], or *Mucor* [23].

When RNAi was discovered, it was thought to be a defense system against exogenous and potentially harmful DNA, including transposons, virus, and transgenes. However, very soon, its involvement in other cellular functions, such as genome integrity or gene regulation, was found. Recent studies in fungal pathogens stated that the RNAi pathway is also implicated in development, drug tolerance, and virulence. Thus, fungi have exploited RNAi to tune their cellular processes, reaching unsuspected limits. The evolutionary relevance of the retention or loss of this regulatory pathway, as well as its diversity of functions in fungi (Table 2), is evaluated in this review.

## 2. Defense against Viruses

The first predicted function of RNAi as a defense against viral infections would explain the conservation of the pathway through the evolution of eukaryotes, since viral infections affect every phylum on the tree of life. RNAi drives virus inactivation through the production of virus-derived small interfering RNAs (vsRNAs) originated from viral dsRNA [70]. This RNAi-based mechanism has been widely explored in the ascomycete filamentous fungus *Cryphonectria parasitica*, which is considered a model for the analysis of virus–fungus interactions because it can be infected by five different virus families. Furthermore, several studies have shown that RNAi regulates virus infection in this fungus [24,25]. *C. parasitica* contains two dicer-like and four Argonaute-like genes. Dcl2 and Ago2 are the primary components of the antiviral defense response, since mutants defective in *dcl2* or *ago2* genes are highly susceptible to mycovirus infection and are affected in the generation of vsRNAs [24,25]. Moreover, there is an induction of *dcl2* and *ago2* expression regulated at the transcriptional level by viral infection [24]. In other fungi, such as *N. crassa*, dsRNA also induces the expression of RNAi genes, supporting the protective role of the RNAi mechanism against virus [71]. On the other hand, viruses have evolved to express RNAi suppressors [72]. The mycovirus *Cryphonectria* hypovirus 1 (CHV1), which can also infect *C. parasitica*, expresses a Papain-like protease. This protein, known as p29, inhibits the expression of *dcl2* and *ago2* in the fungus, thereby avoiding immunity [24].

Furthermore, the mycoviruses that infect the filamentous fungi of the genus *Aspergillus* have been extensively studied, contributing to our understanding of viral diversity and leading to numerous discoveries about virus infection in fungi. In addition to its impact in agriculture and food industry, *Aspergillus* has great relevance in medicine as the causative agent of the fungal infection Aspergillosis. In numerous important *Aspergillus* species, such as *A. nidulans*, *A. fumigatus*, *A. niger*, and *A. flavus*, RNAi seems to be involved in virus–fungus interactions [73]. Specifically, in *A. nidulans*, siRNAs derived from *Aspergillus* virus 341 were detected, indicating that RNAi acts as a defense mechanism against the virus [26]. *A. nidulans* has genes encoding one Argonaute and one Dicer. Although two *rdrp* genes are found in the fungus genome, neither is required for small RNA (sRNA) production, suggesting the absence of silencing amplification [1]. Similarly to viruses that infect *C. parasitica*, infection of *A. nidulans* with *Aspergillus* virus 1816 suppresses RNAi, suggesting the existence of an RNA silencing suppressor encoded by the viral genome [26]. Thus, evolution of the virus–host interaction includes tuning of the host RNAi mechanism for both the virus and the host’s benefit.

## 3. Control of Transposable Elements

Transposable elements (TEs) are described as DNA sequences that have the ability to change their position within a genome. For details on TE classification, we encourage readers to refer to the review by Bourque et al. [74]. Although TEs were initially considered junk DNA, they have been associated with several important activities since their discovery, including centromere function, genome reorganization, and gene expression regulation [75,76,77]. TEs are also considered “selfish” DNA because they try to be perpetuated whilst the host tries to curtail their spread and, thus, their consequences on genome integrity. As a result, many organisms have developed mechanisms to ensure the control of TE activity [78,79]. Some of those mechanisms are well described in the literature and include DNA methylation [80], histone methylation [81], and heterochromatin-inducing protein [82]. Another mechanism that controls TE spread, probably the most ancient of all mentioned, is RNAi. Small RNAs associated with proteins can act at the transcriptional or post-transcriptional level against TE activity. The role of RNAi in TE repression has been well characterized in the plant kingdom [83] and other organisms, such as *Drosophila melanogaster* and *Caenorhabditis elegans* [84,85]. Noteworthy, in *C. elegans* and *Chlamydomonas reinhardtii*, some genes have proven to be essential for both RNAi and TE control mechanisms [86,87,88], while, in mammals, the existence of active siRNA molecules against LINE-1 transposons (Class 1, non-LTR) has been reported [89]. Studies on the RNAi role in TE control in fungi have served to further understand these mechanisms. In *N. crassa*, the repression of LINE1-like retrotransposons is exclusively achieved by post-transcriptional gene silencing and, unlike other eukaryotic species, DNA methylation appears not to be involved [27]. This TE control mechanism shares common elements with quelling (QDE-2 and Dicer) but is independent of QDE-1 or QDE-3, suggesting a diversification of pathways depending on the target. In addition to TE control during mitotic growth, MSUD acts by suppressing repetitive elements during meiosis [19,28]. 

RNAi mechanisms for TE regulation have also been described in several other fungi. Yamanaka et al. [29] reported the existence of siRNAs targeting the Tf2 retrotransposon in *S. pombe,* where a correlation between siRNAs and methylation of histone H3 in lysine 9 (H3K9me) was also detected. In the phytopathogenic fungus *Magnaporthe oryzae,* approximately 10% of all the small RNAs detected in the mycelial stage mapped to repetitive DNA, especially LTR-retrotransposons [30], and RNAi components were proven to be in control of the LTR-retrotransposon MAGGY [31,32]. Previous studies in *M. oryzae* already confirmed that DNA methylation was not implicated in MAGGY repression and suggested the influence of gene silencing [33]. In other fungi with active RNAi machinery, the possible role of RNAi in TE control has also been suggested, i.e., *Aspergillus* [90], or discarded, as occurred in the case of *Trichoderma atroviride* due to the very low number of TEs present on its genome and the absence of small RNAs mapping to these regions [44].

In *Mucor lusitanicus,* formerly known as *Mucor circinelloides* f. *lusitanicus* [91], sRNAs pairing with transposons and derived from the canonical pathway have been detected (see below for a more detailed description of RNAi pathways of this fungus) [43,92]. A recent study discovered the presence of TE sequences surrounding the centromeric area of *M. lusitanicus,* which seem to be present in all Mucoromycotina species and, thus, were termed as Grem-LINE1 (genomic retrotransposable element of Mucoromycota) [34]. The canonical RNAi pathway negatively regulates theses sequences, as mutant strains lacking RNAi typical components (Ago1, Dcl1, and Dcl2) show a decrease in sRNA levels targeting Grem-LINE and, consequently, increased Grem-LINE mRNA expression. Surprisingly, mutant strains lacking proteins (RdRP1 and R3B2) of a different pathway known as noncanonical RNAi pathway (NCRIP) show enhanced levels of Grem-LINE1 sRNAs compared to the wild-type strain [64]. Therefore, the NCRIP regulates the silencing of these Grem-LINE1 through the modulation of the canonical mechanism [64]. 

The pathogenic fungus *Cryptococcus neoformans* has also developed a mechanism to maintain genome integrity by controlling TE activity via RNAi, especially during sexual reproduction. Its sequenced genome revealed the presence of genes encoding the core components of RNAi: one Rdp (Rdp1), two Dicers (Dcr1 and Dcr2), and one Argonaute (Ago1) [93]. Furthermore, a significant accumulation of sRNAs mapping to TE sequences was found in strains undergoing sexual reproduction. However, in a Δ*rdp1* mutant strain, these sRNAs were lacking and a higher expression of retrotransposons was reported [35,36]. Thus, this RNAi mechanism was named sex-induced silencing (SIS) and is significantly more active than silencing during mitotic growth [36]. Moreover, by using the FKBP12-encoding gene *FRR1* as a transposon trap, the Δ*rdp1* mutant strain showed an increased transposition rate during both opposite sex (a-α) and unisexual reproduction (α-α), which evidences the strategic role of SIS as a TE control mechanism [37]. Moreover, the process underlying target sequence recognition in this basidiomycete has been studied in detail. Transcripts targeted by RNAi encode suboptimal splicing signals that result in an abnormally high occupancy of spliceosomes. Then, the mRNA stalled in the spliceosomes is used for siRNAs production by a spliceosome-coupled and nuclear RNAi complex (SCANR), as well as the lariat debranching enzyme (Dbr1) [38]. These siRNAs are then used to silence TE. Therefore, the presence of intron-containing transposon-derived mRNA in those spliceosomes evidences TE silencing.

Recently, a set of five novel genes (*rde1–5*) required for the repression of the DNA transposons *HAR1* was also identified in *C. neoformans* [39]. Rde1 and Rde2 proteins appear to be associated with and located in the nucleolus. Rde1 also associates with Ago1 and Prp43, a helicase involved in disassembling stalled and post-catalytic spliceosomes. Rde4 presents a predicted terminal nucleotidyl transferase domain that resembles polyA polymerases and terminal-uridylyl transferases (TUTases). Since Rde4 is implicated in siRNAs biogenesis, it may be possible for TUTases to have a role of TE control in this fungus, similarly to LINE-1 repression in humans [94]. Interestingly, Rde3, a non-Dicer RNAse III related to *Saccharomyces cerevisiae* Rnt1 protein, could be interacting with Rde5 (function unknown) and with RNA quality control enzymes, such as Rrp6 (component of the nuclear exosome) and Rnh1 (RNase H that cleaves RNA-DNA duplexes) [39]. Moreover, in addition to siRNAs, microRNAs (miRNAs) targeting transposons and pseudogenes have also been detected in *Cryptococcus*, suggesting a possible participation of these molecules in TE control [95]. 

Lastly, analogously to *M. lusitanicus*, centromere sequences in *C. neoformans*, *C. deneoformans,* and *C. deuterogattii* are swamped by transposons [96]. Therefore, by controlling the expression of TEs at centromeres, RNAi could play an important regulatory role in the structural evolution of centromeres. Likewise, centromeres of *Puccinia graminis* f. sp. *tritici* (*Pgt*) are highly repetitive [40]. *Pgt* produces a strong wave of sRNAs derived from centromeric regions in the later stages of infection, probably to maintain genome stability during cell division. These centromeric sRNAs induce silencing of TEs both inside and outside the centromeres, as well as affect neighboring genes [40]. A similar phenomenon occurs in other phytopathogen species, where the surrounding of effector genes (term used when referring to genes implicated in the infection process) by repetitive sequences has been reported. RNAi suppresses not only the transposon sequences but also the effector genes, allowing fungi to avoid detection during infection, as plants deploy intracellular immune receptors involved in the recognition of those effector proteins [97,98]. On the other hand, the presence of transposons flanking effector genes may also confer an evolutive advantage. Due to the mobile nature of TE, the effector genes could be translocated to highly dynamic regions of chromosomes, modifying their regulation and allowing rapid adaptation to changing conditions [99]. 

## 4. Regulation of Endogenous Genes

As described above, the development of RNAi mechanisms represents an evolutive advantage regarding the defense against exogenous nucleic acids. However, specialization of those RNAi mechanisms has led to the establishment of novel post-transcriptional regulation networks of endogenous genes according to the use of Rdp, Dicer, and Argonaute proteins. The human pathogenic fungus *M. lusitanicus*, for instance, shows an intricate RNAi mechanism as a function the interplay of the silencing proteins in three different pathways, named the canonical, epimutational, and noncanonical RNAi pathways. The crosstalk of the RNAi pathways creates a complex network that regulates both basic cellular activities, such as metabolism or vegetative growth, and elaborated mechanisms, including sexual reproduction and pathogenesis [23,100,101]. 

*M. lusitanicus* encodes two Dicer-like (Dcl1-2), three Argonaute (Ago1-3), and three Rdp (RdRP1-3) proteins. RdRP1 is essential for the generation of antisense RNA from transgene transcripts. Triggering dsRNA molecules produced by RdRP1 are then processed by Dcl2 into two different classes of siRNAs, 21 and 25 nt long. Afterward, Ago1 binds to the siRNA molecules and provokes degradation of target transcripts. Lastly, RdRP2 amplifies the silencing signal. This basic RNAi machinery acts as a defense mechanism against exogenous nucleic acid sequences, such as transgenes, viruses, and transposons (see above), but also regulates the expression of endogenous genes through exonic siRNAs (ex-siRNAs), which are produced from exons. This new type of siRNA has a post-transcriptional effect on mRNA [92] and regulates the expression of the genes they are produced from and other genes in trans, with complementary sequences or regulated by the initial targets [41]. Four different classes have been described for the 324 exonic siRNAs found so far, depending on the proteins involved in their biogenesis: class I (Dcl2 and RdRP2), class II (Dcl2 and RdRP1), class III (Dcl1, Dcl2, and RdRP1 or RdRP2), and class IV (Dcl1 and RdRP1 or RdRP2) [92]. The protein Ago1 is implicated in all four classes, for their biogenesis and/or stability [102], and the ribonuclease III, R3B2, participates in the generation of most ex-siRNAs of class II, III, and IV, and some of class I [43]. Nearly 700 genes are regulated by these ex-siRNAs, which are involved in several functions, such as carbohydrate, lipid, or amino-acid metabolism, energy production and conversion, and even defense mechanisms. Thus, ex-siRNAs have an important role in *M. lusitanicus* development and physiological processes [41]. In fact, mutant strains affected in components of the silencing machinery required for biogenesis of ex-siRNAs show reduced asexual sporulation [41,102,103], growth, and altered hyphal morphology [104], and accelerated autolysis as a response to nutrient starvation [102]. Likewise, in *T. atroviride*, sRNAs similar to *M. lusitanicus* ex-siRNAs are essential for growth and development [44]. *Fusarium graminearum* also produces ex-siRNAs, which are involved specifically in the ascospore formation [45]. 

The most recent RNAi mechanism discovered in *M. lusitanicus* is known as NCRIP because none of the known Dicer and Argonaute proteins are involved. A new group of sRNAs, named *rdrp*-dependent *dicer*-independent sRNAs (rdRNAs), was found during the analysis of the double Dicer mutant (Δ*dcl1/dcl2*) of *M. lusitanicus* [43]. The rdRNAs are degradation products of the mRNA without a defined size but with a strong bias for uracil in the penultimate position. Furthermore, they are produced by R3B2, an atypical RNase III-like protein with one RNase III domain and two double-strand RNA-binding domains, together with RdRP1, RdRP3, and the helicase RnhA. Thus, these new sRNAs are not random degradation products of the fungal transcriptome, and NCRIP drives its genetic control through the degradation of the mRNA from its target genes [42,43]. This mechanism was initially described as a gene regulation pathway implicated in oxidative stress and sexual interaction [23,43]. Additionally, the latest studies established that NCRIP regulates more than 25% of the genome, 3187 genes mostly linked to cellular metabolism, germination, and development, indicating a relevant role in saprophytic growth conditions [64]. Furthermore, the transcriptomic profile of *M. lusitanicus* during phagocytosis is similar to that of mutant strains lacking NCRIP [64,101], suggesting that this mechanism is repressing the genetic program of defense against phagocytosis. Thus, the NCRIP also has a relevant role in the pathogenesis of this filamentous fungus (see below for a more detailed description). On the other hand, the epimutational RNAi pathway, which is also reviewed more thoroughly later on, allows in *M. lusitanicus* the transitory silencing of specific mRNA during a stress challenge [42,55,56]. The fact that the inactivation of the NCRIP increments the production of epimutants shows the competition of both RNAi pathways over the messengers with high expression rates [42,43]. This, together with the phagocytosis results, exposes again the relevant role of the NCRIP repressing the defense genetic programs during optimal growth of *M. lusitanicus*. Lastly, the R3B2 protein, which produces rdRNA, is also required by the canonical RNAi pathway of *M. lusitanicus*. As described before, the atypical RNase III is involved in the production of most of the ex-siRNAs, together with RdRP, Dicer, and Argonaute proteins [41,43]. The relationship of the NCRIP with the canonical RNAi and the epimutation pathways reveals the intricate interaction of proteins developed through evolution that allows *M. lusitanicus* refined control over mRNA levels to confront environmental challenges [64,100]. 

On another note, in *N. crassa*, Dicer-independent small interfering RNAs (disiRNAs) emerged as a new type of silencing pathway, without the implication of the classic RNAi proteins for its generation [48]. High levels of disiRNAs result in high levels of DNA methylation and promoter-specific RNA transcripts, suggesting a regulation of endogenous gene expression by DNA methylation, which was, thus, named disiRNA loci DNA methylation (DLDM) [50]. This mechanism has a high dynamism and an on/off pattern, where antisense transcription provokes the stalling of RNA polymerase II and the exonuclease ERI-1 has a main role in both DLDM and disiRNA biogenesis [49]. On the other hand, microRNAs (miRNAs) are small RNAs which are present in many organisms, regulating the expression of endogenous genes by complementarity with their target mRNAs [105]. *N. crassa* produces microRNA-like RNAs (milRNAs), which have significant similarities with miRNAs related to their precursors, their production by Dicer, and their function regulating cellular activities [48]. Different combinations of Dicer proteins, the exonuclease QIP (quelling inducer protein), the RNase III domain-containing protein MRPL3, and the Argonaute QDE-2, are involved in the diverse mechanisms (at least four) that produce these milRNAs [48]. In addition, it has been demonstrated that RNA polymerase II and III participate in their transcription. This remarkable role of RNA polymerase III in the production of milRNAs seems to be one of the differences between milRNAs and miRNAs [106]. milRNAs have also been studied in other organisms. *Coprinopsis cinerea*, a model mushroom, produces milRNAs implicated in development and fruiting body formation [47]; moreover, in *F. graminearum*, milRNAs were initially found not only in asexual phases [107] but also in sexual stages later on [46], where they seem to have an important role in the regulation of sexual development. As described above, *F. graminearum* also produces ex-siRNAs [45]. The participation of Argonaute and Dicer proteins is essential for the biogenesis of both kinds of sRNAs in this fungus, *Fg*Dcl1 and *Fg*Ago2, to generate ex-siRNAs, and *Fg*Dcl1/2 and *Fg*Ago1 to generate milRNAs [45,46]. 

In *Magnaporthe oryzae*, two different sRNAs have been observed: transfer RNA fragments (tRFs) and circular RNAs (circRNAs). tRFs have been detected in bacteria, fungi, plants, and animals [108]. In *M. oryzae*, tRFs were identified in spores and mycelia, but their specific function has not yet been identified [30]. These tRNA fragments are involved in many different cell activities in human and other organisms, including the inhibition of translation [109], repression of endogenous genes involved in cell proliferation and DNA damage response [110], and even the suppression of antiviral responses by a respiratory syncytial virus [111]. tRFs have also been observed in the plant pathogen *Phytophthora infestans*, produced with the implication of *Pi*Ago1, but not *Pi*Dcl1. High levels of tRFs in this fungus are linked to plant infection [52]. On the other hand, circRNAs, which repress miRNAs in animals and plants, were recently discovered in *M. oryzae*. circRNAs have been studied in mycelium and conidium of this fungus, and they are produced from genes involved in metabolism and normal growth, as well as in the biogenesis of lipid, glycogen, and amino-acid storages, which may be used for plant infection. However, circRNAs are still under study, because their general functions are not entirely known [51]. 

Thus, the huge variety of RNAi-based pathways that regulate endogenous gene expression in fungi supports the idea that RNAi appeared early in evolution. RNA silencing may even have played an important role in the differentiation of eukaryotes, as it allows a fine-tuning of gene expression, which would change drastically with few alterations in the implicated proteins.

## 5. Heterochromatin Formation

Heterochromatin constitutes a highly condensed state of DNA. It is considered to have no transcriptional activity due to the limited access of the regulatory proteins to the promoter regions [112]. Generally, heterochromatin is concentrated in the telomeric, centromeric, ribosomal, and mating type regions of the eukaryotic chromosome [113]. Heterochromatin assembly is strictly regulated for accurate chromosome segregation, maintenance of telomere integrity, transcriptional silencing, and transposon control [112,114]. *S. pombe* is an excellent study model to discern and understand the epigenetic mechanism regulating heterochromatin assembly [21]. Furthermore, *S. pombe*’s genome presents a single copy of the genes encoding RNAi proteins: Argonaute (Ago1), Dicer (Dcr1), and RNA-dependent RNA polymerase (Rdp1) [53,115]. Heterochromatin formation in *S. pombe* is triggered by the production of siRNAs derived from centromeric regions with numerous repeats [53,115]. These siRNAs are loaded into the RNA-induced transcriptional silencing complex (RITS) through the Argonaute protein, leading the complex to complementary nascent RNA transcripts from the initial centromeric repeats [116,117]. Chromatin-associated RITS promotes the recruitment of the histone methyltransferase Clr4, a subunit of the cryptic loci regulator complex (CLRC), which is involved in nucleosome methylation and propagation of heterochromatin. The Clr4 enzyme, therefore, catalyzes the methylation of the histone H3 (H3K9me) at the centromeric repeats, inducing heterochromatin assembly [115,117,118,119]. Absence of Clr4 results in the interruption of both events, histone methylation and heterochromatin silencing [117,120]. Additionally, the H3K9me marks at the centromeric regions function as binding sites for chromodomain proteins, such as Chp1 (a component of RITS), Swi6 (a heterochromatin protein 1 homolog), and Clr4 itself. Consequently, H3K9me promotes the recruitment of chromatin-modifying proteins and RNAi components, which stimulate siRNA amplification and the spread of H3K9me domains to adjacent centromeric regions [53,118,121]. Thus, the coupling between RITS/RNAi and CLRC functions is essential for the formation and maintenance of heterochromatin, as well as centromeric silencing in *S. pombe* [121,122]. Nevertheless, heterochromatin formation to maintain epigenetic silencing is a highly complex process and could integrate alternate pathways, as cells lacking the RNAi components maintain basal levels of H3K9me and drive the formation of heterochromatin [123].

Remarkably, heterochromatin formation and the RNAi pathway can also regulate the epigenetic inheritance of gene silencing in *S. pombe* [54]. siRNA amplification derived from hairpin RNA promotes the establishment of the H3K9 methylation at a target euchromatic locus through the recruitment of chromatin-modifying enzymes, leading to heterochromatin formation and gene silencing. Once established, the H3K9me-mediated epigenetic silencing can be stably inherited through multiple generations, even in the absence of the primary RNA [54,119,124]. The H3K9me contained in the parental histones is recognized by chromatin regulatory proteins, which spread the methylation to newly incorporated histones at the target locus [54,117]. Hence, it has been proposed that H3K9me can act as a molecular mark and confer a cellular memory of gene silencing events in past generations [124]. However, this phenomenon only occurs under the same conditions in which the original silencing was acquired. Thus, the epigenetic memory may promote survival of this species in hostile environments [124]. In conclusion, the development of a crosstalk between the RNAi and the H3K9me epigenetic mechanism has been crucial in the evolution of gene silencing in *S. pombe* [54,119].

## 6. Adaptation to Stressful Conditions

The epimutational pathway in *M. lusitanicus* was discovered after the emergence of isolates resistant to the antifungal drug FK506 with no apparent mutations in the target genes [55]. FK506 interacts with the peptidylprolyl isomerase FKBP12, forming a complex that inhibits Calcineurin, a phosphatase activated by calcium-calmodulin necessary for dimorphism and virulence of *M. lusitanicus*. Thus, inhibition of Calcineurin after treatment with this drug enforces *Mucor* to grow as yeast [125]. However, exposure to FK506 produced the emergence of several *Mucor* isolates with mycelial growth. Most of these FK506-resistant isolates harbored mutations in the *fkbA* gene, which encodes FKBP12, or in the *cnaA* or *cnbR* genes, which encode the subunits of Calcineurin, and presented stable drug resistance. However, several FK506-resistant strains did not have mutations in any of those genes. They also showed resistance to rapamycin, which has the same FKBP12 target, but were not resistant to any other tested drug. These FK506-resistant isolates had a complete loss of *fkbA* expression, but *fkbA* levels were restored to wild-type levels, and the isolates became drug-sensitive after several generations of vegetative growth on FK506-free media. Thus, this process was reversible, and it appeared solely epigenetic. The isolates, called epimutants, produced siRNAs from the mature mRNA of *fkbA* gene, with an average of 21–24 nt in length and a prevalence of uracil at the 5′ end. Both are typical features of *Mucor* siRNAs that interact with Argonaute proteins [55,56]. Epimutants developing resistance to other antifungal agents, such as 5-fluoroorotic acid (5-FOA), have also been isolated. In this case, siRNAs were produced from the exonic regions of the *pyrG* or *pyrF* genes, which encode the proteins that transform 5-FOA into a toxic compound for the fungus. Again, the mRNA levels from the target gene were significantly lower in the epimutants than in the wild-type strain [56]. Therefore, the epimutation process does not appear to occur at a specific gene locus, suggesting it might constitute a general mechanism that generates phenotypic plasticity in *Mucor* by silencing key genes and allowing rapid and reversible adaptation to environmental stresses [42,126].

This mechanism of epimutation requires the action of the canonical RNAi pathway, since the deletion of some components of this pathway, such as the Dicer (Dcl1 or Dcl2), Argonaute (Ago1), or RNA-dependent RNA polymerase (RdRP2) proteins, as well as QIP or a Sad-3-like Helicase (RnhA), abolished the ability to produce drug resistance through epimutations [42,55]. On the other hand, deletions of *rdrp1*, *rdrp3,* or *r3b2* genes increased the production of drug-resistant epimutants. Thus, the NCRIP inhibits the mechanism of generation of epimutants [42]. The implication of both RNAi pathways (canonical and NCRIP) in the epimutant process indicates the existence of a balanced regulation of both mechanisms that produces the activation of the RNAi-dependent epimutant pathway under stress conditions or its repression when the mRNA degradation mechanism operates under nonstress conditions [42]. In addition, *Mucor circinelloides* strains (formerly known as *Mucor circinelloides* f. *circinelloides* and a close relative of *M. lusitanicus*) isolated from human patients displayed an overly activated epimutation pathway, suggesting that the selective pressure of the host–pathogen interaction might promote the phenotypic plasticity provided by this mechanism [42,55]. As NCRIP inhibits the generation of epimutation, it may be interesting to analyze if this mechanism is repressed in these strains.

Moreover, after the infection of immunocompetent mice with an epimutant FK506-resistant *Mucor* strain, epimutation was found to be maintained in vivo during infection. However, reversion of epimutation to a wild-type drug-sensitive phenotype was markedly higher in isolates recovered from the brain than from other evaluated organs [127]. Furthermore, in vivo passage of a wild-type strain of *Mucor* led to increased epimutant-driven induction of drug resistance, especially in isolates collected from the brain. This induction of epimutation did not occur during the infection itself but aroused several days after the recovery of the strains from the different organs and under exposure to drug selection. Therefore, the brain environment does not appear to be the direct inducer of epimutation, but rather prepares the fungus to respond quickly to stress, including antifungal drug exposure [127]. *Mucor* is a ubiquitous fungus that can be found mostly in soil, but is also able to infect and grow in both invertebrate and vertebrate hosts. The adaptation to this variety of environments may be favored by this quick change at the epigenetic level [127]. 

## 7. Pathogenesis

RNAi has also been found to play an important role in pathogenesis, more thoroughly studied in plant pathogens. Many crops with worldwide importance are susceptible to being infected by pathogenic fungi, which translates into economic losses. Thus, alternative methods of infection control have been investigated, allowing a deeper understanding of fungal pathogenesis and the involvement of RNAi. Some of those pathogenic fungi, such as *Colletotrichum gloeosporioides* or *M. oryzae,* have active RNAi pathways implicated in growth and conidiation, which influences their pathogenicity. Mutants lacking one or two RNAi genes in those species usually lose their ability to penetrate the leaves [57,58]. Similarly, *Sclerotinia sclerotiorum,* a devastating plant fungal pathogen that causes up to 100% yield losses in crop production and affects a wide array of crops, encodes two *dicer* (*dcr1-2*) and two *argonaute* (*ago2,4*) genes that have been functionally characterized. RNA-silencing-deficient mutants, Δ*ago2* mutant [128] and Δ*dcr1*/*dcr2* double mutant [59], displayed significantly debilitated growth and reduced virulence. Other species, such as *Valsa mali*, which severely damages apple production, and *Fusarium oxysporum* f. sp. *lycopersici,* which causes wilt disease in tomato, show significantly decreased virulence when they lack the Argonaute gene *ago2* [60,61]. In *Penicillium italicum,* a citrus fruit pathogen that generate citrus blue mold, two Dicer genes (*dcr1-2*) have been characterized. Silencing of the *dcr2* gene, but not *dcr1*, incapacitates the fungus to infect [62]. In the same way, the plant fungal pathogen *F. graminearum*, the major causal agent of wheat head blight, encodes two Dicer, two Argonaute, and five Rdp proteins [107]. *Fg*Dcr1 and *Fg*Ago2, which were essential proteins for the generation of ex-siRNAs, are also positive regulators for *F. graminearum* virulence, since mutation of the encoding genes resulted in reduced *Fusarium* head blight development. Moreover, *F. graminearum* produces several mycotoxins that help the fungus to colonize the plant tissues. When the concentration of some of the mycotoxins was analyzed in wheat spikes infected with RNAi mutants, they showed a lower accumulation than in spikes infected with the wild-type, suggesting that the fungal RNAi mechanism may promote *F. graminearum* mycotoxin production [63].

RNAi is also involved in virulence of animal pathogens. In *M. lusitanicus*, knockout mutants in *rdrp1* and *r3b2*, two NCRIP key proteins, showed an augmented oxidative stress tolerance in vitro and a reduced pathogenic potential in vivo, when tested in an immunosuppressed mice model. Analysis of the transcriptomic profiles of NCRIP mutants identified a substantial number of differentially expressed genes in both Δ*rdrp1* and Δ*r3b2* mutants, compared to the wild-type strain. Remarkably, most of the fungal genes regulated by phagocytosis (involved in functions such as cell motility, cytoskeleton, inorganic ion transport, and metabolism) are under the control of NCRIP, suggesting that this RNAi-based mechanism is a master regulator of the response of the pathogen to phagocytosis and, thus, essential for virulence [64].

An important feature of fungal pathogenesis is the mechanism called cross-kingdom RNAi, which has evolved to regulate the host–pathogen interaction. The existence of sRNA trafficking between the host and the pathogen and silencing target genes of the counterparty in trans was first discovered in plants, but afterward extended to mammal systems. Some fungal pathogens have been found to produce sRNAs that function as RNA effectors to suppress host immunity. *Phytophthora sojae,* an oomycete soybean pathogen, produces two effectors, PSR1 and PSR2, that target the host RNAi processes to attenuate the plant’s immune response and to promote the infection. PSR1 inhibits the host biogenesis of both miRNAs and siRNAs, whereas PSR2 specifically impairs siRNAs accumulation [65]. Likewise, *Botrytis cinerea,* a necrotrophic fungal pathogen, produces small RNAs (*Bc*-sRNAs) during infection that hijack the host plant RNAi machinery to induce the silencing of host immunity genes in tomato and the *Arabidopsis thaliana B. cinerea dcl1*/*dcl2* double mutant that could no longer produce *Bc*-sRNAs exhibited reduced virulence, whereas the *A. thaliana ago1* mutant, which lost RNAi function, developed resistance to *B. cinerea* [66]. Thus, both the pathogen and the host’s RNAi mechanisms are crucial for this cross-kingdom phenomenon to work. Similarly, *Puccinia striiformis* f. sp. *tritici* (*Pst*), one of the most destructive pathogens of wheat, produces a microRNA-like RNA 1 (*Pst*-milR1), which suppresses wheat defenses during wheat–*Pst* interaction. Silencing of the *Pst*-milR1 precursor resulted in increased wheat resistance to *Pst* infection [67]. Moreover, in the cereal powdery mildew pathogens *Blumeria graminis* f. sp. *hordei* (*Bgh*) and *Blumeria graminis* f. sp. *tritici* (*Bgt*), sRNA-seq data derived from infected wheat revealed six putative sRNAs from *Bgt* and 15 putative sRNAs from *Bgh*. Those sRNAs had predicted targets exclusively present in plants, with associated functions that would alter primary metabolism, suggesting potential cross-kingdom RNA transference [68]. Lastly, in the oomycete biotrophic pathogen *Hyaloperonospora arabidopsidis,* 34 sRNAs with the capacity to translocate into plant cells and suppress host target genes were found. However, *Arabidopsis* siRNA biogenesis mutant strains displayed increased *H. arabidopsidis* growth, demonstrating the important role of siRNAs in plant immunity [69]. Therefore, the host–pathogen interaction is constantly evolving, and RNAi has become an advantage for both the host and the pathogen. A fragile balance will decide which succeeds, infection or immunity.

## 8. Loss of RNAi

In essence, RNAi has crucial regulatory and defense roles in eukaryotes, suggesting that this key mechanism has been positively selected through evolution in plants, nematodes, animals, and fungi. Yet, some members of the fungal kingdom have lost key components of the RNAi pathway [129], resulting in its inactivation (Table 3). A hypothesis to explain this contradiction could be that those species may have other defensive mechanisms more advantageous than RNAi. Alternatively, perhaps, this RNA-based mechanism constitutes a disadvantage for them, forcing the survival of the RNAi-deficient species. Loss of RNAi affects species of the subphyla Ustilaginomycotina, Saccharomycotina, Wallemiomycetes, and some members of the phylum Microsporidia [130]. Those organisms lack key components of the RNAi machinery: Dicer, Argonaute, and Rdp proteins. *Ustilago maydis,* for instance, has lost genes encoding the main components of this RNA degradation pathway, even when related species such as *Ustilago hordei* have a functional RNAi mechanism [131]. Furthermore, genes involved in heterochromatin formation, DNA methylation, and repeat-induced point mutation (RIP), a mechanism that inhibits TE activity, are also found in *U. hordei* but have no homologs in *U. maydis* [131]. Therefore, this fungus lacks the usual mechanisms to maintain genome integrity. The absence of typical defense mechanisms might be partially compensated by its very active recombination system, which could facilitate genome stability [131]. Moreover, a correlation between loss of RNAi and length of centromeres has been observed among *Ustilago* and *Cryptococcus* species [96]. While the centromeric regions of the RNA-proficient species *C. neoformans* and *C. deneoformans* are composed of full-length transposable elements [96], *C. deuterogattii,* deficient in RNAi, lacks full-length TE sequences and, thus, features a shorter centromeric region [96,132]. Similarly, *U. maydis* harbors shorter centromeres and fewer transposons than *U. hordei* [96]. Those findings establish a relationship between RNAi and centromere structure evolution. On the other hand, the loss of RNAi in *C. deuterogattii* is likely associated with the evolution of an increased virulence since free movement of retrotransposable elements could provide rapid adaptative responses to the stressful environment host cells entail through a major genome plasticity [132]. 

Lastly, some budding yeast, including the human pathogen *Candida albicans* and *Naumovozyma castellii* (previously known as *Saccharomyces castellii* and close relative of *S. cerevisiae*), have lost some components of the RNAi machinery, but have replaced them with noncanonical RNAi proteins to produce dsRNA and short hairpin RNA (shRNA) as regulatory and defensive elements against transposable elements [133]. However, *S. cerevisiae* lacks the key RNAi components and, thus, the ability of TE inhibition by RNAi. To understand why RNAi has been negatively selected in these organisms, we should take into account their evolutionary context. For example, *S. cerevisiae* can be infected by mycoviruses known as “Killer viruses”, which are dsRNA viruses that can be cytoplasmically inherited. This endemic virus encodes for a toxin that kills the neighboring cells lacking the “Killer system” (not infected by the mycovirus), while conferring immunity to those making the toxin. Restoring the RNAi mechanism in *S. cerevisiae,* by introducing the Dicer and Ago proteins from the related species *N. castellii,* contributes to genome stability through TE silencing [133], but negatively affects the Killer system. The acquirement and retention of Killer virus, thus, favored the evolution of the RNAi-deficient species [134]. Therefore, the role and even the presence of RNAi differ across all fungal kingdom, as it has evolved adapting to the particularities and needs of each species. Fungi have exploited RNAi to conquer different niches, adapt to diverse stresses, acquire drug tolerance, and avoid or even modulate the host immune response during infection. According to these RNAi-mediated roles, it would not be wrong to say that RNAi has played a very significant role in the evolutionary success of fungi as a kingdom, as well as individual organisms.

## Figures and Tables

**Table 1 ijms-21-09348-t001:** RNA interference (RNAi) mechanisms in eukaryotes.

Eukaryotic Group	Evolutionary Advantage	Loss of RNAi	Effector Proteins	siRNAs	References
Fungi	Defense against viruses Control of TE Regulation of endogenous genes Heterochromatin formation Adaptation to stressful conditions Pathogenesis	*Saccharomyces cerevisiae* *Ustilago maydis* *Cryptococcus deuterogatti*	Ago Dicer Rdp/RdRP	siRNA masiRNA milRNA ex-siRNA endo-siRNA rdRNA circRNA tRF disiRNA	Table 2 and Table 3
Protists	Defense against viruses Control of TE Heterochromatin formation DNA elimination	*Trypanosoma cruzi* *Leishmania major* *Leishmania donovani* *Leishmania tarentolae* *Plasmodium falciparum*	Piwi Dicer RdRP	siRNA scnRNA	[3,4,5,6,7]
Nematodes	Defense against viruses Control of TE Heterochromatin formation Regulation of endogenous genes		Ago/Piwi Dicer Rrf/Ego	miRNA piRNA endo-siRNA exo-siRNA	[3,5,8]
Insects	Defense against viruses Control of TE Heterochromatin formation Regulation of endogenous genes		Ago/Piwi Dicer/Loquacious/ Drosha/Pasha/R2D2	siRNA miRNA piRNA endo-siRNA	[3,5,8,9,10]
Plants	Defense against pathogens Control of TE Heterochromatin formation Regulation of endogenous genes Repair DNA double-strand breaks (DSB)		Ago Dcl Rdr	siRNA miRNA hp-siRNA hc-siRNA or hetsiRNA tasiRNA t-siRNA lsiRNA nat-siRNA phasiRNA easiRNA diRNA	[3,8,11,12,13]
Mammals	Defense against viruses Control of TE Heterochromatin formation Regulation of endogenous genes Repair DNA double-strand breaks		Ago/Piwi Dicer/Drosha	siRNA miRNA piRNA endo-siRNA diRNA	[3,5,8,13,14]

TE: transposable elements; Ago: Argonaute; Rdp/RdRP/Rdr: RNA-dependent RNA polymerase; Dcl: Dicer-like protein; siRNA: small interfering RNA; masiRNA: MSUD-associated siRNAs; MSUD: meiotic silencing of unpaired DNA; milRNA: microRNA-like RNAs; ex-siRNA: exonic siRNAs; endo-siRNA: endogenous siRNA; rdRNA: *rdrp*-dependent *dicer*-independent sRNAs; circRNA: circular RNAs; tRF: transfer RNA fragments; disiRNA: Dicer-independent siRNAs; scnRNA: small scan RNA; miRNA: micro RNA; piRNA: Piwi-interacting RNA; exo-siRNA: exogenously-derived siRNA; hp-siRNA: hairpin-derived siRNA; hc-siRNA or hetsiRNA: heterochromatic siRNA; tasiRNA: trans-acting siRNA; t-siRNA: transgene-derived siRNA; lsiRNA: long siRNA; nat-siRNA: natural antisense siRNA; phasiRNA: phased siRNA; easiRNA: epigenetically-activated siRNA; diRNA: DSB-induced siRNA.

**Table 2 ijms-21-09348-t002:** Classification of RNA interference (RNAi) mechanisms found in fungi.

Function	Regulation of	Main RNAi Proteins	siRNAs	RNAi Mechanism	Fungi Species	References
**Defense against viruses**	*Cryphonectria* hypovirus 1	Dcl2, Ago2	vsRNAs		*Cryphonectria parasitica*	[24,25]
	*Aspergillus* virus 1816	DclB, RsdA	vsRNAs		*Aspergillus nidulans*	[26]
**Control of Transposable Element**	LINE1-like retrotransposons	QDE-2, Dcl1, Dcl2		Quelling	*Neurospora crassa*	[27]
	DNA transposon *Sly1-1*	Dcl1, SAD-1-5, SMS-2	masiRNAs	MSUD	*Neurospora crassa*	[19,28]
	Tf2 retrotransposon	Ago1, Rdp1, Dcr1, Clr4		RNA-induced transcriptional silencing (RITS)	*Schizosaccharomyces pombe*	[29]
	LTR-retrotransposon MAGGY	*Mo*Dcl2, *Mo*Ago1, *Mo*Ago3	LTR-siRNAs		*Magnaporthe oryzae*	[30,31,32,33]
	Grem-LINE1	Ago2, Dcl1, Dcl2		Canonical	*Mucor lusitanicus*	[34]
	Retrotransposons	Rdp1, Dcr1, Dcr2, Ago1, Qip1, Gwc1, Srr1	endo-siRNAs	Sex-induced silencing (SIS) Spliceosome-Coupled And Nuclear RNAi complex (SCANR)	*Cryptococcus neoformans*	[35,36,37,38]
	DNA transposon *HAR1*	Rde1-5	endo-siRNAs		*Cryptococcus neoformans*	[39]
	Young TE	Ago1, Dicer3, RdRP1/3	Late wave sRNAs		*Puccinia graminis* f. sp. *triticci*	[40]
**Regulation of endogenous genes**		RdRP1, RdRP2, Dcl1, Dcl2, Ago1, R3B2	ex-siRNAs	Canonical	*Mucor lusitanicus*	[23,41]
	Canonical RNAi, phagocytosis response	RdRP1, RdRP3, R3B2, RnhA	rdRNAs	Non-canonical (NCRIP)	*Mucor lusitanicus*	[42,43]
	Growth and development	Dcr2, Rdr3	ex-siRNAs		*Trichoderma atroviride*	[44]
	Ascospore formation	*Fg*Dcl1, *Fg*Ago2	ex-siRNAs		*Fusarium graminearum*	[45]
	Sexual development	*Fg*Dcl1/2, *Fg*Ago1	milRNAs		*Fusarium graminearum*	[46]
	Development and fruiting body formation		milRNAs		*Coprinopsis cinerea*	[47]
		Dicer, QIP, MRPL3, QDE-2	milRNAs		*Neurospora crassa*	[48]
		ERI-1, QDE-2	disiRNAs	disiRNA loci DNA methylation (DLDM)	*Neurospora crassa*	[49,50]
	miRNA		circRNAs		*Magnaporthe oryzae*	[51]
			tRFs		*Magnaporthe oryzae*	[30]
	Plant infection	*Pi*Ago1	tRFs		*Phytophthora infestans*	[52]
**Heterochromatin formation**	Centromeric regions	Dcr1, Ago1, Rdp1, Tas3, Chp1	Centromeric siRNAs		*Schizosaccharomyces pombe*	[53,54]
**Adaptation to stressful conditions**	FKBP12 PyrG, PyrF	Dcr1-2, Ago1, RdRP2, QIP, RnhA		Epimutation	*Mucor circinelloides* *Mucor lusitanicus*	[55,56]
**Pathogenesis**	Ability to infect leaves	Dcr1, Dcr2			*Colletotrichum gloeosporioides*	[57]
	Ability to infect leaves	Ago3, Rdp1			*Magnaporthe* *oryzae*	[58]
	Virulence	Dcr1-2, Ago2			*Sclerotinia sclerotiorum*	[59]
	Resistance to H_2_O_2_ and virulence	Ago2			*Valsa mali*	[60]
	Virulence	Ago2			*Fusarium oxysporum* f. sp. *lycopersici*	[61]
	Infection	Dcr2			*Penicillium italicum*	[62]
	Infection	*Fg*Dcl1, *Fg*Ago2			*Fusarium graminearum*	[63]
	Resistance to H_2_O_2_ and virulence	RdRP1, R3B2	rdRNAs	NCRIP	*Mucor lusitanicus*	[64]
	Host miRNA and siRNA		PSR1 and PSR2	Cross-kingdom RNAi	*Phytophthora sojae*	[65]
	Host immunity genes		*Bc*-sRNAs	Cross-kingdom RNAi	*Botrytis cinerea*	[66]
	Host defenses		*Pst*-milR1	Cross-kingdom RNAi	*Puccinia striiformis* f. sp*. tritici*	[67]
	Host metabolism genes		*Bgh*-sRNAs *Bgt*-sRNAs	Cross-kingdom RNAi	*Blumeria graminis* f. sp. *hordei* *Blumeria graminis* f. sp. *tritici*	[68]
	Host genes			Cross-kingdom RNAi	*Hyaloperonospora arabidopsidis*	[69]

Grem-LINE1: genomic retrotransposable element of Mucoromycota; TE: Transposable Elements; miRNA: micro RNA; siRNA: small interfering RNA; Dcr: Dicer; Dcl: Dicer-like protein; Ago: Argonaute; Rdp/RdRP/Rdr: RNA-dependent RNA polymerase; vsRNAs: virus-derived siRNAs; masiRNA: MSUD-associated siRNAs; MSUD: meiotic silencing of unpaired DNA; endo-siRNA: endogenous siRNA; ex-siRNA: exonic siRNAs; rdRNA: *rdrp*-dependent *dicer*-independent sRNAs; milRNA: microRNA-like RNAs; disiRNA: Dicer-independent siRNAs; circRNA: circular RNAs; tRF: transfer RNA fragments; sRNAs: small RNAs; NCRIP: noncanonical RNAi pathway.

**Table 3 ijms-21-09348-t003:** Loss of the RNAi mechanism among fungi species.

Species	Putative Advantage	Lost Proteins	Substitutes	References
*Ustilago maydis*	Retention of Killer virus	Ago1, RdRP1-3, Dcr1	Partially by recombination system	[131]
*Crytpococcus deuterogatti*	Increased virulence by TE activity	Ago1, Ago2, Dcr1, Rdp1		[96]
*Candida albicans*		Dicer	Noncanonical *Ca*Dcr1	[133]
*Naumovozyma castellii*		Dicer	Noncanonical *Nca*Dcr1	[133]
*Saccharomyces cerevisiae*	Retention of Killer virus	Ago1, Dcr1		[134]

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
