# Peer review of "The Evolutionary Significance of RNAi in the Fungal Kingdom"

_ijms, 2020, doi:10.3390/ijms21249348_

Round 1

Reviewer 1 Report

Lax et al., review RNA interference strategies in the fungal kingdom. In principle, they succeeded very well and I have only several comments to illuminate their knowledge even better:

  1. The Table comparing eukaryotic and fungi components of RNAi is missing.
  2. The Table indicating missing components of RNAi and potential replacement strategies is missing (to the last chapter “Loss of RNAi”).

Author Response

We really appreciate the kind words and the support. We have created the suggested tables:

  1. This is now Table 1, at the bottom of the first page of the revised manuscript, entitled “RNAi mechanisms in Eukaryotes”. We think this table is not really necessary as it includes a lot of information not covered in the text. Since the review is only about fungi, we do not think it is relevant to include all these data. But we do not have any problem if the editor decides to include it in the manuscript. It may be useful to a broader audience, thus, it may be worth it to include it.
  1. Table 3 “Loss of the RNAi mechanism among fungi species” covers the missing components in the different fungal species, the putative advantage of the loss, and possible substitutes when appropriate. The new table is in page 14. We really like this one, it is a nice summary of the section.

Reviewer 2 Report

RNA interference is silencing mechanism centered on processing of dsRNA molecules generated by RNA-dependent RNA polymerase and processed by RNAse III (Dicer) into small RNA molecules that are utilized to guide Argonaut-mediated degradation of the targeted RNA. RNAi is often organized in positive feedback loop that stimulates conversion of targeted transcript into dsRNA. Small RNA molecules produced by Dicer can form epigenetic memory. In the submitted manuscript authors provide comprehensive overview of RNAi in fungal kingdom, and describe RNAi involvement in various cellular functions, including gene regulation, antiviral defense, suppression of transposable elements, adaptation to stressful conditions and heterochromatin formation.

The manuscript by Carlos Lax et al has provided detailed review of RNAi activity and significance across the fungal kingdom. The work is appropriate for Int. J. of Mol. Sci.  The review is well organized, the argumentation and language of the manuscript are clear.

Comments:

1] Main comment:

I am concern about the selection of cited papers. In heterochromatin section the high-profile papers that should have been cited for the role of RNAi in heterochromatin targeting are omitted:

Thomas Volpe et al (2002) Regulation of heterochromatic silencing and histone H3 lysine-9 methylation by RNAi. Science, vol 297, pp 1833-1837

Ira M. Hall et al (2002) Establishment and maintenance of a heterochromatin domain. Science vol 297, pp 2232-2237

Andre Verdel et al (2004) RNAi-mediated targeting of heterochromatin by the RITS complex. Science vol 303, pp 672-676

2] Minor comments:

I believe that naming the RNAi-mediated resistance to drug FK506 in M. lusitanicus epimutation is not appropriate; this is surly adaptation to drug that needs to be positively selected, however the resistance is rapidly lost after drug removal. Epimutations as observed in plants are stable in absence of selection and can be transmitted even through meiosis.

Author Response

We are really grateful for the kind review, and we are thrilled you liked the manuscript. Please find below the answers to your comments:

1) You are completely right. We included those citations in the first draft of the review. They must have been lost somewhere in the editing process. We tried to update the bibliography to include the most recent papers, but still keeping the essential studies and first reports in each section. We apologize for this major overlook. The three documents have been included again among the citations:

Volpe et al, 2002: citation 115 in line 308, 310, and 316

Verdel et al, 2004: citation 116 in line 312

Hall et al, 2002: citation 120 in line 322 and 323

2) Epimutations in Mucor are stable for more than 20 generations in some isolates in absence of selection, while lost in 5-10 in others. We were not able to check if it would be conserved through meiosis because Zygospores from Mucor do not germinate in lab conditions. We named this phenomenon as "epimutations" in 2014, when it was published in Nature. Since then, several other high-profile papers have been published by Heitman's team. I understand you may not like the term used in this specific situation, but since it has been called that way for so long, I don't think a review of RNAi in fungi is the place to change it. I am sure Joseph Heitman will be thrilled to discuss with you if the term is applicable or not.